# A scoping review to map the concept, content, and outcome of wilderness programs for childhood cancer survivors

**Mats Jong**[1]*, **E. Anne Lown**[2], **Winnie Schats**[3], **Michelle L. Mills**[2], **Heather R. Otto**[4], **Leiv E. Gabrielsen**[5], **Miek C. Jong**[1,6]

**1** Department of Health Sciences, Mid Sweden University, Sundsvall, Sweden, **2** Department of Social and Behavioral Sciences and Osher Center for Integrative Medicine, University of California San Francisco, San Francisco, California, United States of America, **3** Scientific Information Service, Netherlands Cancer Institute, Amsterdam, The Netherlands, **4** See you at the Summit, Portland, OR, United States of America, **5** Department for Child and Adolescent Mental Health, Sørlandet Hospital Health Enterprise, Kristiansand, Norway, **6** Department of Community Medicine, Faculty of Health Sciences, National Research Center in Complementary and Alternative Medicine (NAFKAM), The Arctic University of Norway (UiT), Tromsø, Norway

* mats.jong@miun.se

**Data Availability Statement:** All relevant data are within the manuscript and its Supporting Information files.

## Abstract

### Objectives

Systematic mapping of the concept, content, and outcome of wilderness programs for childhood cancer survivors.

### Design

Scoping review.

### Search strategy

Searches were performed in 13 databases and the grey literature. Included studies describe participation of childhood cancer survivors in wilderness programs where the role of nature had a contextual and therapeutic premise. At least two authors independently performed screening, data extraction and analysis.

### Results

Database searches yielded 1848 articles, of which 15 met the inclusion criteria. The majority of programs (73%) employed adventure therapy. Five activity categories were identified as components of wilderness programs: challenge/risk, free time/leisure, experiential learning, physical activity and psychotherapeutic activities. A majority of the participating childhood cancer survivors were female, white, aged 8–40 years, with a wide range of cancer diagnoses. Reported outcomes included increased social involvement, self-esteem, self-confidence, self-efficacy, social support, and physical activity. Key gaps identified included the absence of randomized controlled trials (RCTs), lack of studies on long-term effects, lack of information on the multicultural aspects of programs, and missing information on engagement in nature activities after the program ended.

**Funding:** MJ and MCJ were supported by internal research funding of the Department of Health Sciences at Mid Sweden University, Sweden (www.miun.se). This research received no specific grant from any other funding agency in the public, commercial or not-for-profit sectors. The funder had no role in study design, data collection and analysis, decision to publish, or preparation of the manuscript.

**Competing interests:** The authors have declared that no competing interests exist.

## Conclusions

This scoping review guides childhood cancer survivors, their families, practitioners, clinicians and researchers in the development and optimization of wilderness programs for childhood cancer survivors. In addition, it informs the utilization of these programs, and identifies gaps in the evidence base of wilderness programs. It is recommended that future study reporting on wilderness programs include more detail and explicitly address the role of nature in the program. Performing RCTs on wilderness programs is challenging, as they occur in real-life contexts in which participants cannot be blinded. Creative solutions in the design of pragmatic trials and mixed method studies are thus needed for further investigation of the effectiveness and safety of wilderness programs in childhood cancer survivors.

## Introduction

Due to major advances in diagnosis and increased use of multimodal treatment, childhood cancer survival has significantly improved in recent decades. The overall five-year survival rate for childhood cancer has grown from 58% in the mid-1970s to more than 80% [1, 2]. However, survival rates vary significantly by cancer type, and there remains significant global inequality in survivorship [3]. As survival rates have improved, increasing attention has been placed on the long-term health problems and associated needs of childhood cancer survivors. Many survivors suffer from late health effects as a consequence of the cancer and cancer-related treatments. Late effects in childhood cancer survivors may be psychosocial/behavioral related, such as depression, anxiety and risky health behaviors, or physical related, such as cardiovascular disease, secondary malignancies, and hormone and immune deficiencies [4, 5]. Many childhood cancer survivors thus carry a significant risk of late morbidity and mortality [6, 7]. Health-promoting interventions aimed at assisting and supporting them in order to promote optimal quality of life in light of these difficulties are therefore needed.

Health promotion is defined as action directed toward the support of active and healthy living and facilitation of supportive environments for health [8]. Time in nature has gained renewed interest as an affordable health promotion strategy that can enhance mental, physical and social well-being [9, 10]. Wilderness therapy is an example of a health promotion strategy where the nature-human interaction is applied in a therapeutic context. It is used with a wide range of populations, but most commonly used with adolescents at risk for mental, behavioral and/or drug-related problems [11]. There is no consistent or universally accepted definition of wilderness therapy, which makes it difficult to compare and replicate the outcome of studies on wilderness therapy. In current literature, wilderness therapy is identified interchangeably with numerous other terms such as challenge courses, adventure-based therapy, wilderness experience programs, nature therapy, therapeutic camping, recreation therapy, outdoor therapy, open-air therapy and adventure camps [12, 13]. Together with this extensive variability in terminology, there are multiple definitions of wilderness therapy One definition advocated by Russell defines wilderness therapy as an intervention that utilizes outdoor adventure activities, such as primitive skills and reflection, to enhance personal and interpersonal growth [13]. Davis-Berman describes wilderness therapy as a group treatment modality in mental health care that seeks to augment the restorative qualities of nature in combination with structured and intentional individual and group-based therapeutic work [14]. Fernee et al. propose that wilderness therapy distinguishes itself from the larger group of wilderness experience

programs in that it encompasses elements specifically targeted toward the treatment of adolescent emotional, behavioral, psychological, and/or substance use issues [15]. The different uses of terminology and variability in definition of wilderness therapy reflect the historical and socio-cultural contexts and traditions in which these programs have developed. In the US, youth camping programs and experiential education are commonly regarded as the predecessors of wilderness therapy [14]. In countries like Canada and Australia, wilderness programs are rooted in the traditional travel and living practices of Indigenous people and early European explorers and settlers [16]. In the Scandinavian countries, wilderness therapy developed from the "friluftsliv" (outdoor life) tradition in which a deep affiliation with nature and the simple life outdoors are essential [17]. These different approaches to wilderness therapy share a basic theoretical underpinning grounded in ecology, ecosophy and deep ecology, whereby the health and wellbeing of the natural world is intrinsically interwoven in a bi-directional fashion with the health and wellbeing of humans [18, 19]. The role of nature in wilderness therapy is mostly explained and supported in evolutionary and biological terms [20], but has also been criticized for lack of a more in-depth exploration of the human-nature relationship in reporting of outcomes [21, 22]. Several systematic reviews/meta-analyses have been carried out that explored the effects of nature-based programs [23, 24], adventure therapy [25, 26], wilderness therapy [27, 28] or cancer camps [29, 30]. An extensive systematic review that included 461 articles on nature-based experiences reported a wide range of benefits related to mental, physical, and social health [23] across different populations. A narrative review focused on nature-based experiences of cancer survivors concluded that being in nature supports quality of life, sense of belonging, and self-esteem, and in addition decreases state anxiety [24]. A meta-analysis on the effects of adventure therapy demonstrated moderate effects in facilitation of short-term change in psychological, behavioral, emotional and interpersonal domains for any given (patient) population [25]. Another systematic review reported that little to no difference was found between the different therapeutic factors in adventure therapy [26]. A meta-analysis of wilderness therapy outcomes for private pay clients demonstrated medium-sized effects for its clients within the areas of self-esteem, locus of control, health behaviors, personal effectiveness, clinical symptomology, and interpersonal skills [27]. A previous scoping review on wilderness therapy identified a need to more clearly identify and articulate outdoor adventure practices as they relate to child and youth care practice [28]. In addition, two systematic reviews have reported that participation in pediatric cancer camps positively affects the social health, self-concept, quality of life, sense of normalcy and emotional well-being of childhood cancer survivors [29, 30]. Cancer camps aim to provide children with a "normal" vacation in which they can participate in a range of recreational activities, and where they have the opportunity to play and share experiences with peers in a fun, stress-free environment [31]. Although cancer camps often take place in an outdoor or wilderness setting, these and other previously published systematic reviews have not specifically addressed or investigated the role of wilderness and nature in such programs for childhood cancer survivors. Therefore, the present scoping review was initiated with the aim of systematically mapping the concept, content and outcome of wilderness programs for childhood cancer survivors [32]. In the absence of a consistent and universally accepted definition, a pragmatic operational concept of wilderness programs was predefined for the purpose of this scoping review as follows: "Wilderness-related therapies such as adventure therapy, recreation programs, nature-based programs, outdoor programs, open-air programs, forest bathing and bushcraft, in which the role of nature has both a contextual and therapeutic premise" [32]. The role of nature in these programs required that a wilderness program takes place in nature, and that nature-related activities within the program be intended to have a therapeutic effect [16, 33]. The purpose of this scoping review was to inform childhood cancer survivors, their families, practitioners, clinicians, and

## Methods

The original protocol for this scoping review was published in 2019 [32]. The protocol followed the Joanna Briggs Institute (JBI) Reviewers' Manual for scoping reviews [34] and guidance for conducting systematic scoping reviews as published by Peters et al. [35]. Given the objectives and related research questions of this study, a scoping review approach was deemed the most suitable type of review method. The purpose of a scoping review is to scope a body of literature in order to clarify key concepts and definitions, identify key characteristics related to that concept, examine how research is conducted on that topic, identify knowledge gaps, and identify the types of available evidence [36]. Results are reported according to the Preferred Reporting Items for Systematic review and Meta-Analyses extension for Scoping Reviews (PRISMA-ScR) checklist [37] (S1 File). This scoping review analyzed data already published in the literature. Therefore, the present study was exempt from medical ethical review.

### Objectives

The aim of this review was to map the concept, content, and outcome of wilderness programs for childhood cancer survivors and addressed the following questions:

**Review question 1.** What concepts of wilderness programs (e.g. theoretical frameworks, foundations) are presented for childhood cancer survivors?

**Review question 2.** Which elements (content) are incorporated into wilderness programs for childhood cancer survivors (e.g. experiential learning methods, physical movement, challenge and risk-based activity, the generating and use of metaphors, involvement with natural environments, balance of structured and unstructured time in the program, balance of social and individual time in the program, and different type of habitats and habitat-specific activities) and which elements have not been incorporated for childhood cancer, but may be promising?

**Review question 3.** Which professionals (e.g. profession, qualifications) facilitate wilderness programs for childhood cancer survivors, and what relationship have wilderness programs had with treatment institutions?

**Review question 4.** What benefits and risks (outcomes) are reported for wilderness programs in childhood cancer survivors?

**Review question 5.** To what extent are elements of the wilderness encounter incorporated into the daily life of childhood cancer survivors, how is this incorporation influenced by their domestic situation, and how does that benefit their health in the longer-term?

**Review question 6.** At what stage of treatment or survivorship are wilderness programs offered to childhood cancer survivors?

**Review question 7.** What is the age range of childhood cancer survivors engaging in wilderness programs?

**Review question 8.** To what extent does the socio-economic situation of childhood cancer survivors affect their participation in wilderness programs, and their continued ability to engage with nature/wilderness after the program?

**Review question 9.** To what extent do disabilities (including physical, sensory or intellectual impairments) of childhood cancer survivors affect their participation in wilderness programs, and their continued ability to engage with nature/wilderness after the program?

**Review question 10.** What is the methodological quality of the included studies on wilderness programs for childhood cancer survivors?

Review question 11.   What are the key gaps in literature around wilderness programs for childhood cancer survivors?

Review question 12.   Are there any ethical issues or challenges identified that relate to participation of childhood cancer survivors in wilderness programs?

## Eligibility criteria

Included studies:

- Involved childhood cancer survivors, including children, adolescents, and young adults who have had a diagnosis of childhood cancer before the age of 21. An individual was defined as a cancer survivor from the moment of cancer diagnosis throughout life [38];

- Focused on wilderness-related programs such as adventure therapy, recreation programs, and other programs in which the role of nature has both a contextual and therapeutic premise;

- Were research articles that used quantitative and/or qualitative methodology, including studies published as master's or bachelor's theses;

- Were reported in the English, Swedish, Norwegian, German or Dutch language.

For inclusion, studies had to minimally describe: 1. a wilderness program targeted towards childhood cancer survivors; 2. a description of the content of the wilderness program; and 3. at least one reported (health-related) quantitative or qualitative outcome. Studies that lacked one or two of these were excluded. Studies focusing on related topics that did not primarily evaluate wilderness and/or nature experiences, such as evaluation of hospital gardens, physical exercise programs, and animal-assisted therapy, or which did not explicitly offer a program (e.g. individuals spending time hiking or star gazing on their own) were also excluded from this scoping review [32].

## Search strategy and study selection

Searches were performed by one reviewer (WS) between 23 April 2019 and 24 May 2019 in the following databases: CINAHL, Cochrane Library, EMBASE, ERIC, Google Scholar, Medline (Ovid), Psycinfo, Scopus, Sociological Abstracts, SPORTDiscus, and Web of Science. A second reviewer (MJ) performed additional searches in that period in AMED and Svemed+ (see S2 File for the results of the comprehensive search strategy). Searches were not restricted to any study design, date or language. Medical Subject Headings (MeSH) (or comparable controlled vocabularies) and free text terms (usually terms in title and/or abstract of the publications) were used in databases with controlled vocabulary. In databases without controlled vocabularies, the search strategy was translated as broadly as possible to have maximum search yield. The detailed MeSH terms and free text/title/abstract terms for the search strategy and the sources for the additional grey literature search have been published previously [32]. One reviewer (MLM) performed searches in grey literature sources. The retrieved records were uploaded in Endnote to facilitate the study selection process and duplicates were removed. Two reviewers (MJ, MCJ) first independently screened titles and abstracts for eligibility and classified the articles into one of the following groups:

1. Not about wilderness programs,

2. Wilderness programs other than for cancer,

3. Wilderness programs for adult cancer survivors,

4. Wilderness programs for childhood cancer survivors.

Initial disagreements in classification of articles were resolved by further discussion between the two reviewers, obviating the need for consultation with a third author. The full text PDFs of articles in category 2–4 were uploaded in Endnote, and one reviewer (MJ) and two other reviewers (EAL, MCJ, who each screened about half of the articles) independently screened full texts of all articles for eligibility and inclusion in the scoping review. The three reviewers (MJ, EAL, MCJ) discussed whether the eligibility criteria were met for each included article. In addition, two reviewers (MJ, MCJ) independently screened full reference lists of all articles in category 2–4. Additional articles, identified through screening of reference lists, underwent the same process of screening, data extraction and data synthesis as the other articles. Since the selected articles retrieved through the searches addressed the research questions, it was not deemed necessary to broaden the search terms and eligibility criteria for inclusion of other articles into the scoping review. Because the authors of this review had an excellent understanding of the five languages described in the inclusion criteria, translation of articles in this scoping review was not necessary.

## Data extraction

Two reviewers (MCJ, WS) independently performed a pilot data extraction of one article [39]. After piloting, the authors assessed the extracted data in relation to the scoping review questions. Accordingly, four items were added to the data extraction form (S3 File). The two reviewers (MCJ, WS) continued data extraction of all articles independently (see S1 Dataset). Differences in data extraction were discussed and resolved between the two reviewers, and missing items were identified.

## Quality appraisal

Study quality assessment is not usually a part of a scoping review [35]. In the present scoping review however, a quality appraisal was performed in order to evaluate the quality of studies and identify possible gaps in literature (but not for reasons to identify studies for exclusion). Therefore, included articles were exported to the System for the Unified Management, Assessment and Review of Information (SUMARI software program, JBI 2019) for critical appraisal. Two reviewers (MCJ, EAL) independently rated the methodological quality of included articles using the critical appraisal checklists in SUMARI. Since SUMARI does not yet contain a critical appraisal checklist for mixed-methods studies, the quality of included articles with a mixed-methods design were assessed using the Mixed Methods Appraisal Tool (MMAT) [40]. Discrepancies between the reviewer's quality assessments were discussed and resolved. For the purpose of this review, articles with $\geq$ 75% positive (yes) score on the critical appraisal items were classified to be of high quality, from 50–74% of medium quality, and $<$ 50% of low quality.

## Collating and summarizing the results

Four reviewers (MCJ, MJ, WS, EAL) were involved in the process of data synthesis and interpretation. Where numerical data of specific study characteristics were available, descriptive statistics were used to summarize the data. Otherwise, a narrative approach was used to collate and summarize the data.

## Deviations from the protocol

Rather than assessing the quality of qualitative research articles using the Critical Appraisal Skills Program [41] as per the published protocol [32], the JBI checklist for qualitative studies

in SUMARI [42] was used. Furthermore, since JBI does not have a critical appraisal tool for mixed-methods studies, the quality of the articles that used a mixed-methods design was assessed with the MMAT [40]. Another deviation from the published protocol was that the term "wilderness programs" replaces the term "wilderness therapy" in the title and objective. In the absence of a universally accepted definition of wilderness therapy, the authors deemed it more appropriate to use the more generic term "wilderness programs" in relation to the purpose of this scoping review.

### Patient and public involvement

Patient representatives of Young Cancer Sweden (Ung Cancer: https://ungcancer.se/) participated in the reporting of results. Data interpretation and reporting was supported by two wilderness therapy practitioners (HRO and LEG), both having more than 10 years' working experience with children and adolescents in the outdoors.

## Results

### Article identification and selection process

The database and grey literature search yielded 1848 records after deduplication (Fig 1). Screening of titles and abstracts resulted in a first classification, after which 1744 records were excluded because they were not on wilderness programs. After full-text screening of the resulting 104 records, 30 were excluded because they were not on (childhood) cancer, and 24 records that were identified through screening of reference lists were added. A total of 94 records were therefore assessed for eligibility, of which 15 articles met the eligibility criteria for inclusion in the scoping review. The listing of the three possible relevant articles that were excluded because of language can be found in S4 File.

### Characteristics of included articles

A summary of the included articles and their study designs (n = 15) [39, 43–56] can be found in S5 File. Included articles were methodologically diverse, consisting of text & opinion papers (n = 5) [43–46, 49], qualitative studies (n = 4) [51–54], studies using a mixed-methods design (n = 3) [47, 48, 55], or quasi-experimental studies (n = 3) [39, 50, 56]. Most articles originated in the USA (n = 10) [39, 43–45, 47, 49, 50, 53, 54, 56], others were from Canada (n = 3) [46, 48, 52], the UK (n = 1) [51], and New Zealand (n = 1) [55]. Four out of 15 articles were published prior to 2000 [43, 45, 47, 49]. In most articles, participants were adolescent cancer survivors (n = 7) [43, 45, 46, 48, 49, 52, 55], followed by young adult cancer survivors (n = 5) [39, 50, 51, 53, 56], or children with life-threatening/chronic diseases including cancer (n = 2) [44, 47]. One article describes the experiences of adolescent cancer survivors through the eyes of recreational therapists [54].

### Concepts and goals of wilderness programs

**Review question 1.** The majority of articles (73%, 11 out of 15) concerned adventure therapy. Of the remaining four articles, two were about wilderness experience, one about cancer camp, and one involved a ski-rehabilitation program (see S5 File). Most articles included scant detail on the theoretical concepts behind their programs. Overall, the articles conveyed that through mastery, performance-based success, and use of metaphorical associations, programs aim to empower participants by increasing self-worth, inner strength, confidence, and self-esteem. In addition, they described that these program experiences can transfer to behavioural and attitudinal changes in the survivor's everyday life [44, 46, 47, 49, 51–54]. Underlying

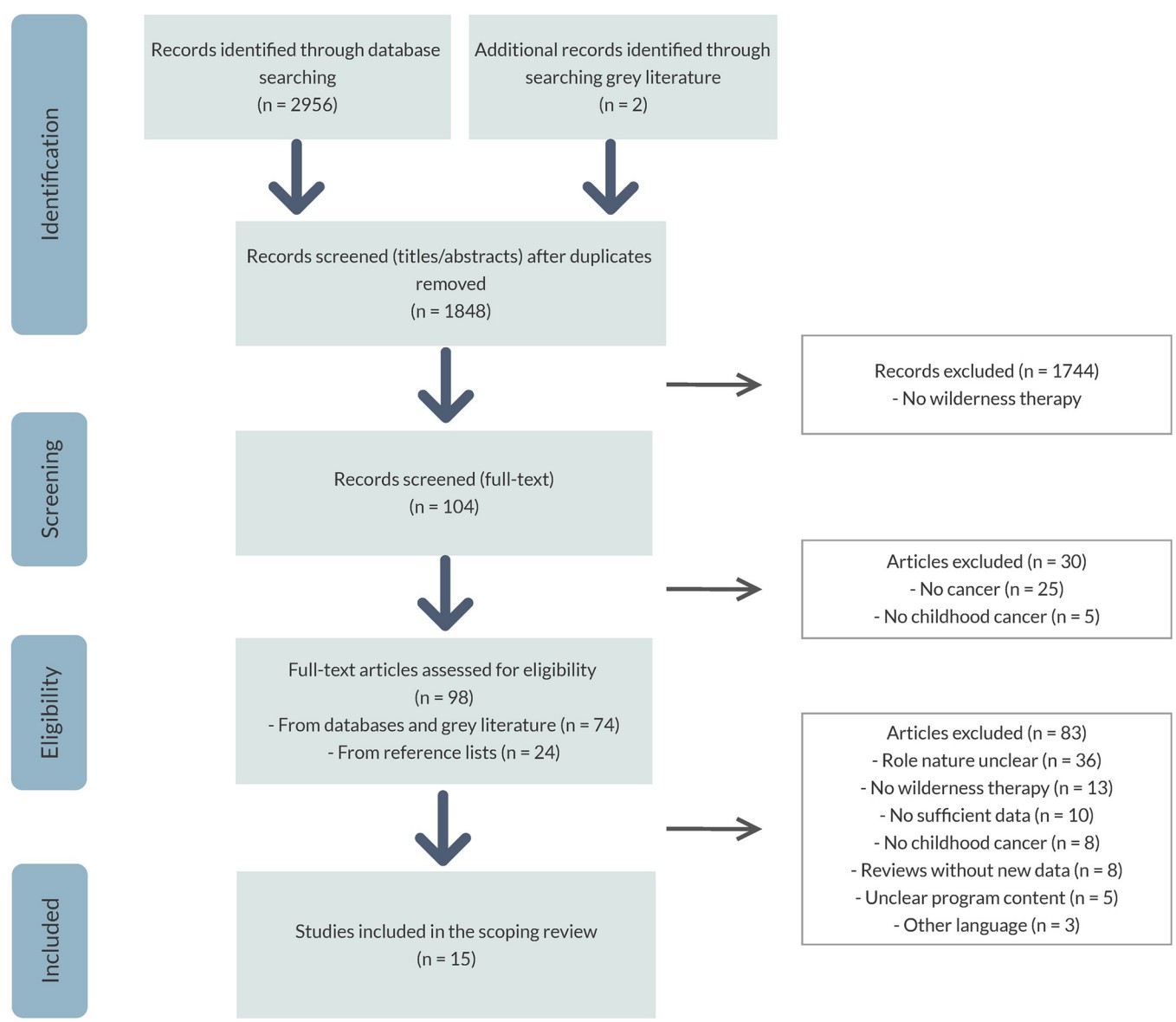

**Fig 1. PRISMA flow diagram.**

theoretical foundations cited in some of the articles [47, 48, 55], were self-determination theory, experiential learning, and development principles [57, 58]. In six of the articles, the programs aimed to meet developmental needs of participants, thereby encouraging personal growth and providing participants with new tools to support them as to move forward in their lives [43, 46, 49, 51, 52, 55]. In practical terms, the type of programming included: 1) outdoor activities involving (physical) challenge; 2) a nature/wilderness setting; and 3) the supportive environment of facilitators and peers. In the case of the cancer camp, the primary aim was to provide participants with a normal camping experience in an environment where they could derive benefit interacting with peers having similar life challenges and experiences, and where they could take part in everyday activities so that they do not feel different from others [45].

While all articles reported nature/wilderness as the environment in which their program took place and included some kind of description from which it could be concluded that

nature had a therapeutic intention in the program, most did not describe in much detail on the role of nature within their program. Nature was described as having positive and restorative effects [39, 46, 52, 56] as it simplifies life [52], breaks down inappropriate defenses and denial [52], gives peace and comfort [53], offers a feeling of freedom or an expression of spirituality [46], and allows participants time to reflect and get in touch with themselves [46, 52].

## Content and characteristics of wilderness programs

**Review question 2.** The content of programs is shown in Fig 2. A wide variety of program activities was grouped into five categories: 1) challenge/risk activities such as kayaking; 2) free time/leisure activities such as using mobile devices; 3) experiential learning activities such as map and compass orienting; 4) physical activity such as hiking; and 5). psychotherapeutic work such as the use of metaphors. Activities in the challenge/risk category were mostly reported in adventure therapy, while activities in the free time/leisure category were mostly reported in the cancer camp. Activities in the wilderness experience and ski-rehabilitation

### Challenge/risk activities

| Activity | Adventure Therapy | Wilderness Experience | Cancer Camp | Ski-Rehab |
|---|---|---|---|---|
| Abseiling | X | | | |
| Dog sledding | X | | | |
| Ghyll scrambling | X | | | |
| Hill walking & mountaineering | X | | | |
| (White water) kayaking | X | | | |
| Midnight mine walk | X | | | |
| River rafting | | X | | |
| Rock/mountain climbing | X | X | X | |
| Rope courses | X | X | | |
| Sailing | X | | | |
| Seaplane ride | X | | X | |
| (Sit) skiing | X | X | | X |
| Snowmobiling | X | | | X |
| Snowshoeing | X | | | |
| Surfing | X | | | |
| Tubing | | | | X |
| Tyrolean traverse | X | | | |
| Wilderness survival | X | | | |

### Free time/leisure activities

| Activity | Adventure Therapy | Wilderness Experience | Cancer Camp | Ski-Rehab |
|---|---|---|---|---|
| Amusement park | | | X | |
| Boat ride | X | | X | |
| Camp fire | X | X | X | X |
| Clown visits | | | X | |
| Dancing | | | X | X |
| Fire truck ride | | | X | |
| Fire works | | | X | |
| Flag raising | | | X | |
| Hay ride | | | | X |
| Mountain chair lifts | | | X | |
| Museum visit | | | | |
| Music | X | | | |
| Pizza party | | | | X |
| Singing | | | X | X |
| Snowball fights | | | | X |
| Talent show | | | X | |
| Using mobile devices | X | | | |
| Watching TV/Video | X | | | |

### Experiential learning

| Activity | Adventure Therapy | Wilderness Experience | Cancer Camp | Ski-Rehab |
|---|---|---|---|---|
| Air gun rifle | | | X | |
| Archery | X | | X | |
| Arts & crafts | X | | X | X |
| Camping | X | X | X | |
| Cooking out | | X | X | |
| Creative writing | X | | | |
| Equipment planning | | X | | |
| Fishing | X | | X | |
| Map and compass orienting | | X | | |
| Photography | X | | | |
| Theater | X | | | |
| Safety skills training | | X | | |
| Videography | X | | | |

### Physical movement

| Activity | Adventure Therapy | Wilderness Experience | Cancer Camp | Ski-Rehab |
|---|---|---|---|---|
| Backpacking | | X | | |
| Canoeing, portaging | X | X | X | |
| Hiking | X | X | | |
| Horseback riding | X | | X | |
| Swimming/diving | X | | X | X |
| Trips in nature | X | | X | |

### Psychotherapeutic work

| Activity | Adventure Therapy | Wilderness Experience | Cancer Camp | Ski-Rehab |
|---|---|---|---|---|
| Journaling | X | | | |
| Problem-solving games | X | | | |
| Reflective exercises | X | | | |
| Relay games | | | X | X |
| Use of metaphors | X | | | |

**Fig 2. Content of wilderness programs.** This figure depicts the variety of program activities (X) that were included in adventure therapy, wilderness experience, cancer camp or in the ski-rehabilitation program.

program were more equally distributed among the categories, although no activities were reported in the psychotherapeutic work category for the wilderness experience program (Fig 2).

Additional characteristics of programs are shown in Table 1. All or nearly all programs were camp-based, took place in nature, and had a closed group structure with facilitators and participants being together from the beginning to the end of the program. The program length varied between 3–14 days, and the group size between 6–11 participants. Only the cancer camp reported a group size between 50–100 participants [45]. Few articles reported on the time spent in group- or structured activities or on the number of facilitators per group.

Activities not listed in Fig 2 that may be of interest to future wilderness programs because of their reported positive effects on mental health in childhood cancer survivors include mindfulness [59–61], yoga [62], and bringing along a dog in the program [63].

**Review question 3.** As shown in Table 1, a broad variety of professional facilitators were involved in the programs, among them health care professionals such as nurses and physicians, outdoor guides, child life workers, and counsellors. At least six out of 15 articles were written by nurses, and included a description on the perspective, relevance, or role of nurses in wilderness programs [43, 45, 46, 49, 52, 55]. Former patients, volunteers or family members were also reported to facilitate in the programs (Table 1). In seven out of 15 articles (47%), it was reported that a treatment institution had initiated the program [43, 45, 47, 49, 52, 54, 55]. Two studies reported that medical clearance from the treating physicians was required for participation in the program [39, 52].

## Outcomes, safety and transference of wilderness programs

**Review question 4.** Fig 3 shows all health-related outcomes that increased upon participation in a wilderness program. According to the definition of health from the World Health Organization (WHO) [64], outcomes were grouped in the domain of mental health (light blue outcomes), social health (orange outcomes), and physical health (dark blue outcomes) (Fig 3). The most commonly reported health benefits were an increase in social involvement [45, 48, 52–54], self-esteem [48, 50, 52, 53, 55], self-confidence [44, 45, 48, 54], self-efficacy [44, 47, 51, 56], social support [51, 53, 55, 56], and physical activity [39, 47, 48, 51] (S6 File). Health-related outcomes that decreased upon participation in a wilderness program included discomfort [48, 50], psychological distress [48, 56], and alienation [48, 50] (S6 File).

Only two out of 15 articles (13%) reported on the safety of wilderness programs [43, 45]. Side-effects of participation in the cancer camp were insect bites, abrasions, sore throats, headaches, upset stomachs, sliver in the finger, and laceration of the head of a facilitator [45]. Minor injuries and giving up skiing because of anxiety, were reported for participation in the ski-rehabilitation program [43].

**Review question 5.** Four studies reported that activities such as running [51], skiing [43], recreational activities [47], or other unspecified physical activity [39] were incorporated to a greater extent in the daily life of childhood cancer survivors after participation in the program. Wagner reported that transference, the process by which lessons learned during the program are incorporated into life after the program, was the most reported value that the interviewed participants took from the program [53]. This transference was measured either directly [53] or six months [47] after participation in the program, and included being inspired to stay healthy after camp, recognizing the importance of relationships, being inspired to connect with family after camp [53], or increased social involvement outside the family after the program [47]. Longer-term (> six months) investigation of how this transference affects the health of childhood cancer survivors was not undertaken.

**Table 1. Characteristics of wilderness programs.**

| Characteristics | Percentage* (n) | References |
|---|---|---|
| Program type | | |
| Adventure therapy | 73% (11) | [39, 44, 46, 48, 50–56] |
| Wilderness experience | 13% (2) | [47, 49] |
| Cancer camp | 7% (1) | [45] |
| Ski-rehabilitation | 7% (1) | [43] |
| Program model | | |
| Camp | 47% (7) | [39, 43–45, 49, 51, 53] |
| Expedition | 27% (4) | [48, 52, 54, 55] |
| Not reported | 27% (4) | [46, 47, 50, 56] |
| Setting | | |
| National park/wilderness/nature | 100% (15) | [39, 43–56] |
| Program length | | |
| 3–5 days | 13% (2) | [51, 54] |
| 6–8 days | 40% (6) | [39, 43, 50, 53, 55, 56] |
| 10–14 days | 27% (4) | [45, 47, 48, 52] |
| Not reported | 20% (3) | [44, 46, 49] |
| Group size | | |
| 6–11 participants | 40% (6) | [39, 43, 44, 47, 52, 53] |
| 50–100 participants | 7% (1) | [45] |
| Not reported | 53% (8) | [46, 48–51, 54–56] |
| Group structure | | |
| Closed group | 93% (14) | [39, 43–45, 47–56] |
| Open group | 0% (0) | |
| Not reported | 7% (1) | [46] |
| Amount of time in group activities | | |
| Minimum 4 hours | 7% (1) | [44] |
| Not reported | 93% (14) | [39, 43, 45–56] |
| Amount of time in structured activities | | |
| 5–7 hours | 7% (1) | [39] |
| Not reported | 93% (14) | [43–56] |
| Number of facilitators per group | | |
| 2–5 facilitators | 33% (5) | [43, 44, 47, 52, 55] |
| Not reported | 67% (10) | [39, 45, 46, 48–51, 53–56] |
| Type of facilitators in the program** | | |
| Health care professionals | 53% (8) | [43–47, 51, 52, 55] |
| Professional (outdoor) guides | 33% (5) | [39, 46, 47, 50, 51] |
| Program staff | 27% (4) | [39, 44, 50, 55] |
| Child life educators/counsellors | 27% (4) | [43–45, 52] |
| Former patients/ambassadors | 20% (3) | [43, 45, 51] |
| Volunteers | 7% (1) | [43] |
| Family members | 7% (1) | [43] |
| Not reported | 33% (5) | [48, 49, 53, 54, 56] |

*Percentages may be more or less than 100%, due to rounding.

** More categories per program is possible

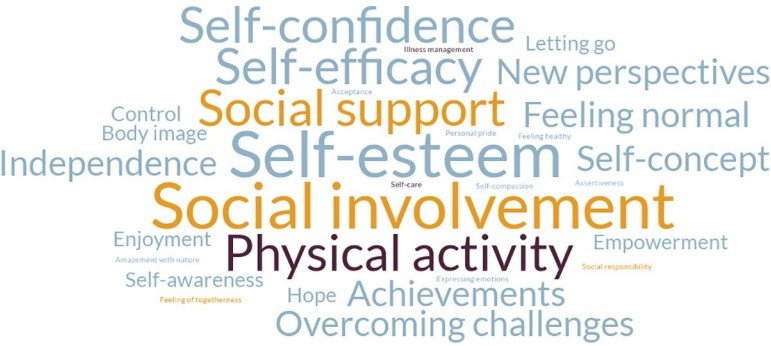

**Fig 3. Increase in health-related outcomes of wilderness programs.** Mental health (light blue), social health (orange), physical health (dark blue).

## Characteristics of childhood cancer survivors

**Review question 6.** Four articles reported the time between cancer diagnosis/cancer treatment and participation in the program, which was on average 0–4.5 years (n = 345) [39, 43, 50, 53]. As shown in Table 2, participants had a variety of cancer types and previous cancer treatments, and about half of them (47%) were on active treatment (medication/therapies) for cancer or late effects of cancer at the time of their participation in the program.

**Review question 7.** The articles described cancer survivors (n = 1383) with a broad age range from age 8–40 years (n = 1350) [39, 45, 47, 48, 50–53, 55, 56]. As shown in Table 2, the

Table 2. Characteristics of childhood cancer survivors in the articles.

| Characteristics | Percentage (n) | References |
|---|---|---|
| Gender (n = 740) | | [39, 48, 50–53, 55, 56] |
| Female | 78% (606) | |
| Male | 22% (175) | |
| Ethnicity (n = 478) | | [39, 53, 56] |
| White | 86% (410) | |
| Non-white (Hispanic, Asian, Other) | 14% (68) | |
| Domestic situation (n = 483) | | [39, 53, 56] |
| Married/in relationship | 36% (174) | |
| Not married/not in relationship/divorced | 64% (309) | |
| Socio-economic situation (n = 416) | | [39, 56] |
| Work/school | 74% (306) | |
| No work/no school | 26% (110) | |
| Cancer diagnosis (n = 160) | | [39, 43, 47, 51, 52] |
| Breast | 24% (39) | |
| Non-Hodgkin/Hodgkin | 18% (30) | |
| Leukemia | 7% (11) | |
| Brain tumor | 6% (9) | |
| Thyroid | 5% (8) | |
| Other | 39% (63) | |
| Previous cancer-treatment (n = 401) | | [39, 43, 50] |
| Chemotherapy | 30% (122) | |
| Radiation | 27% (109) | |
| Surgery | 42% (170) | |
| Active medication/therapies (n = 345) | 47% (162) | [39, 43, 50] |

majority of childhood cancer survivors were female (78%), white (86%), not married or in a relationship (64%), and working or in school (74%).

**Review question 8.** Although only four articles specifically reported that program participation was without cost for childhood cancer survivors [39, 43, 53, 55], it appeared from other sources (such as websites) that the programs in the included articles were free of charge for childhood cancer survivors. No information was reported on how their socio-economic situation affected their ability to further engage in nature/wilderness activities after the program.

**Review question 9.** Participation in wilderness programs was possible for childhood cancer survivors with various comorbidities and/or disabilities, impairments, or late health effects caused either by the cancer itself or by cancer-related treatment. This included mobility, memory and cognitive impairments, amputation, processing difficulties, vision and hearing impairments, seizure disorders, weakness/fatigue and balance problems, hormonal deficiencies and nocturnal enuresis, or special individual treatment or dietary needs [39, 43, 44, 50, 51, 55]. It was not reported how these programs were specifically adapted to meet these needs, or how these disabilities affected participants' continued engagement with nature/wilderness activities after the program.

## Methodological quality of studies

**Review question 10.** Of the 15 articles included, ten articles were published in scientific journals [39, 43–47, 49, 50, 52, 56], two were master's theses [51, 54], one a bachelor's thesis [53], one was published as a symposium proceeding [48], and one was published in a national nursing journal [55]. Twelve out of 15 articles (80%) were assessed at high or medium quality. Eight articles of high quality scored positive (yes) on most or all quality items in the critical appraisal tools (S7 File) [39, 44, 46, 50–52, 54, 56]. Another four articles were of medium quality, of which three text & opinion articles scored positive on 4 out of 6 quality items [43, 45, 49], and one qualitative study scored positive on 6 out of 10 quality items [53]. The three articles with a quasi-experimental design were of high quality and scored positive on all quality items with the exception of one study that did not include a control group [56]. Three out of four articles with a qualitative study design [52–54] did not include a statement that located the researchers culturally or theoretically, nor did they include a description on the influence of the researcher on the research or vice versa. The text & opinion articles suffered from some methodological flaws. Three out of five of these articles [43, 45, 49] did not refer to the extant literature, nor did they discuss any incongruence with the literature. All three articles of mixed-methods design were assessed to be of low quality, scoring negative (no) on ≥ 4 out of 7 quality items (S7 File) [47, 48, 55]. None of these three articles gave an adequate rationale for using a mixed-methods design, nor did they describe the divergences and inconsistencies between quantitative and qualitative findings in the studies.

## Key gaps in literature

**Review question 11.** Upon analysis of the articles on wilderness programs for childhood cancer survivors, the following key gaps were identified:

- Lack of Randomized Controlled Trials (RCTs) on effectiveness: None of the articles in the scoping review had a randomized controlled study design. Two out of 15 studies recommended that future RCTs are needed with larger study samples [39, 51];

- Lack of information on possible safety issues related to wilderness programs for childhood cancer survivors: Only two out of 15 articles reported on side effects of participation in wilderness programs [43, 45];

- Lack of information on long-term (> six months) effects of wilderness programs: Out of the 15 included studies, only one qualitative study [51] and one mixed-methods study [48] reported on long-term outcomes (from one to three years after the program);

- Lack of information on the multicultural and multi-lingual nature of these programs: Aside from one article that recommended inclusion of translation facilitation during the expeditions for those participants with a multicultural background [48], no information on this topic was provided;

- Lack of information on how socioeconomic status and disability affect participants' continued engagement with nature/wilderness activities after the program: None of the articles addressed these issues.

### Ethical issues

**Review question 12.** Since participation appeared to be without cost for childhood cancer survivors, wilderness programs seem to rely heavily on external funding, philanthropic support, and sponsorship. One ethical issue identified in this scoping review pointed to the need for caution regarding inclusion of mass media in the program [48]. Sponsor-driven media presence may interfere with the implementation of the intervention and negatively affect the lived experiences of childhood cancer survivors in wilderness programs.

Another important issue that was uncovered is that childhood cancer survivors who participate in these types of wilderness programs are predominantly white. Furthermore, it was reported that in general they seem to be more physically active and less isolated compared to peers [39, 50, 56]. A major challenge for these programs is thus is how to reach out to and recruit racially and ethnically diverse childhood cancer survivors, as well as to survivors who are less physically active and more isolated.

## Discussion

### Main findings

This scoping review provides valuable and new insights into the concept, content and outcome of wilderness programs specifically targeted towards childhood cancer survivors. The majority of programs (73%) in this scoping review were about adventure therapy. Analysis of included program content identified five categories of activities: challenge/risk, free time/leisure, experiential learning, physical activity and psychotherapeutic activities. Although all programs had in common placement in a wilderness/nature setting in which nature had a therapeutic premise, it was shown that they differed in the types of activities they provided. For example, adventure therapy focused mostly on challenge/risk activities in nature, whereas the cancer camp included many free time/leisure activities in their program.

Reported health- and psychosocial-related outcomes that increased after participation in wilderness programs included social involvement, self-esteem, self-confidence, self-efficacy, social support, and physical activity. Health-related outcomes that decreased upon participation in a wilderness program included discomfort and psychological distress. It seems unlikely that the observed decrease in discomfort and psychological distress in the studies were the result of higher anxiety levels and discomfort at baseline as caused by the prospect of taking part in an unfamiliar wilderness program. That distress is usually understood by survivors as transitory and arising briefly in relation to the trip, and not the longer-term psychological distress that has been demonstrated among childhood cancer survivors compared with sibling controls [65]. Second, in the study by Rosenberg et al. [50], baseline measurements were

performed two weeks prior to the start of the program, and discomfort after participation was significantly decreased in the outdoor adventure program group compared to a wait-list control group. Third, in the study by Zebrack et al. [56], baseline measurements were also performed two weeks prior to the start of the program. In this study, psychological distress directly after participation in the program decreased significantly more in those with moderate-to-severe distress at baseline compared to those with mild-to-none. However, psychological distress increased again at follow-up one month after participation in the program [56].

Although only two studies in this scoping review reported on the safety of wilderness programs, side effects in these studies appeared to be minor [43, 45]. A study that monitored health incidents in wilderness therapy programs for youth with mental health problems demonstrated that participation in these programs is safer than either playing high school football or daily life at home [66]. The potential health benefits of these programs for childhood cancer survivors thus appears to outweigh possible risks. The observed positive impact of wilderness programs on social and mental well-being of childhood cancer survivors in this scoping review is in line with the outcome of two previously published systematic reviews on the impact of cancer camps [29, 30]. A third systematic review on cancer camps by Kelada et al. [67] was recently published and also demonstrated that cancer camps, in addition to increasing social and mental well-being, may positively impact physical well-being of childhood cancer survivors. However, Kelada et al [67] also reported that the evidence for health benefits of cancer camps was mixed. Some of the articles in their review reported non-significant changes in health-related outcomes, as well as the occurrence of tension between families of children on treatment and bereaved families [67]. Social, mental and physical health benefits of wilderness programs have also been reported for youth with mental health and behavioral issues, and for those with substance abuse diagnoses [25, 27, 68].

Childhood cancer survivors participating in wilderness programs represented a broad age range and a broad variety of cancer diagnoses, similar to participants in cancer camps [29, 30, 67]. This scoping review revealed that participants were predominantly white and female, an observation that was not discussed in previous systematic reviews on cancer camps. This is in direct contrast to wilderness programs for at-risk youth, where the majority of participants were male [25, 27, 68]. The female overrepresentation in the identified studies cannot solely be explained by the cancer incidence of the age groups in the population since childhood cancer is slightly more common among boys than girls [69, 70], and boys (0–19 years) also have a higher incidence of mortality [70]. However, young women (20–39 years) have a disproportionally higher cancer incidence as well as mortality than men [71, 72].

Although the majority (80%) of included articles in this scoping review were of medium to high quality, there is an apparent lack of RCTs, high quality case studies, and mixed-methods studies on the effectiveness of wilderness programs. Systematic quality assessment of studies on cancer camps reported similar conclusions with respect to the limitation of studies due to small sample sizes and lack of multisite, longitudinal, and controlled study designs [67].

## Strengths and limitations

Strengths of this scoping review include *a priori* establishment and publication of the review protocol [32], as well as the fact that at least two authors independently performed screening, data extraction and analysis. Other strengths include the international nature of the research team that carried out this review (Sweden, USA, Netherlands, Norway), and the inclusion of expertise from the fields of epidemiology, pediatric psycho-oncology, academia, clinical medicine, wilderness therapy, and scientific information services, and the inclusion of childhood cancer survivors' and wilderness therapy practitioners' perspectives on interpretation and dissemination of the results.

A limitation is the fact that the 15 articles included were heterogeneous in terms of programs, aims, and methodology, although this is in line with the characteristics of a scoping review. The fact that wilderness programs were operationalized as programs in which the role of nature has both a contextual and therapeutic premise resulted in the inclusion of four types of programs, among them a cancer camp that reported nature activities as one of its components [45]. Many additional articles on cancer camps have been published, but these were not included due to the fact that they did not explicitly mention (the role of) nature in their program [29, 30]. While nature is most often integrated into camp life, lack of explicit mention of nature may have excluded articles that could be relevant for childhood cancer survivors. On the other hand, it can be argued that a wilderness program/wilderness therapy is more than "merely" a camp in nature; rather, it could be seen as essential that considerable time be spent embedded in wilderness without any interference from people outside of the group [12, 16]. Five out of the 15 articles (33%) in this scoping review met this "remote wilderness" criterion [47–49, 52, 55]. The heterogeneity of articles as included in this scoping review may thus reflect existing differences in definitions, and in the historical and socio-cultural contexts and traditions in which wilderness programs have developed [14, 73]. Another limitation of this scoping review is that it included four articles on young adult cancer survivors [39, 50, 53, 56], where the age at cancer diagnosis was not specified. It is thus not certain that the diagnosis occurred before the age of 21, and the results reported in these articles quite likely do not specifically target childhood cancer survivors, but rather young adult survivors in general who may have differing needs [74]. Furthermore, two included articles reported on seriously ill children [44, 47] (not all of whom were cancer survivors), and one article focused on the experiences of recreational therapists [54], which may not fully represent the perspectives of childhood cancer survivors.

## Implication for practice

The results of this scoping review are of interest to a broad audience including childhood cancer survivors, their families, practitioners, clinicians, and researchers.

This scoping review provides childhood cancer survivors and their families with information about the specific activities that are part of wilderness programs, and also reassurance that any current or future comorbidities and/or disabilities do not have to restrict participation in such a program. It also provides them with information on the safety of the programs, as well as information on potential health benefits survivors may gain through participation. Although participation in wilderness programs appears to be free of cost to the participants in the articles included in this scoping review, it has been reported that funding and third party reimbursement for outdoor programs have declined and that more programs need to be privately supported [75]. Childhood cancer survivors and their parents should therefore always inquire about possible program-related costs before signing up to participate. In countries such as Norway, where the healthcare system is publicly funded, programs are in principle cost-free [76]. This usually requires the intervention in question to be evidence based and supported by a clear and unambiguous rationale for why it should be offered. However, when participation in wilderness programs solely relies on private funding, there is a greater likelihood that these types of programs will enroll participants with greater economic privilege and exclude those with lower socioeconomic status. To the best of our knowledge, no one has studied this issue.

The detailed information provided by this scoping review on the concepts, content, group size, facilitators and other characteristics of wilderness programs will be of interest to practitioners interested in developing or improving their own wilderness program. Based on the

findings presented here, it is recommended that practitioners explicitly develop the role of nature within the program. In this way, 'being in nature/wilderness' can be activated consciously and used to further well-being/reduce stress (as has been reported in previous studies [77–79]), rather than simply being a background setting or a stage for activities focusing on overcoming physical and mental challenges [22]. Wilderness could serve as a source of solace or spiritual and existential meaning, allowing childhood cancer survivors time to ground, reflect, and touch in with themselves, thereby enhancing restoration and transformation [11, 52]. It is thus recommended that in addition to the five program-activity categories identified in this scoping review, a sixth program-activity category that specifically integrates the role of nature in wilderness programs be added in order to encourage introspection, awareness, gratitude and personal growth. For instance this could include mindfulness-based activities related to the beauty and grandeur of nature such as forest bathing [80], open air relaxation breathing, walking meditations, or yoga [60]. It is also recommended that practitioners work toward finding ways to attract a more diverse group of participants, including those from multi-ethnic and varied socioeconomic backgrounds.

Most articles included in this scoping review refer to wilderness programs as therapies. This aligns with the current trend in healthcare whereby nature and exercise are increasingly being "medicalized" into interventions such as nature on prescription and physical activity on prescription [81–83]. However, it is advisable to frame wilderness programs either as therapies or as health promotion strategies, depending on the program content and needs of the target group. Childhood cancer survivors for example often have prolonged and varied cancer treatment-related therapies, and may therefore not be so willing to get involved in 'yet another' therapy.

Last but not least, this scoping review identified several challenges for clinicians and researchers with an interest in building the evidence base of wilderness programs for childhood cancer survivors. First, clinicians/researchers need to improve reporting in studies on wilderness programs and childhood cancer survivors. Articles included in this scoping review lacked information on at least five or more study characteristics that were extracted for the purpose of this review. Details about the theoretical concepts behind the program, the content, the group structure, facilitators, and socio-economic status of participants and other program and participant characteristics are crucial pieces of information that further guide program development and research. In addition, most articles did not report on the safety of wilderness programs for childhood cancer survivors. Monitoring and reporting on possible safety issues related to wilderness programs such as incidents, injuries, altitude sickness, medical conditions, physical exertion (carrying backpacks), hostile relationships, and medication use in participating childhood cancer survivors is strongly recommended.

Second, further research is needed to investigate how wilderness program activities can be effectively incorporated into the daily lives of childhood cancer survivors, and whether this affects their health in the longer term. Third, there is a need for high-quality case studies and in-depth qualitative/mixed method studies to build and deepen the conceptual understanding of how and why different wilderness programs might be influencing different aspects of health and wellbeing for childhood cancer survivors, and how this might inform the design of future intervention studies. Textbooks on wilderness therapy connect theoretically with the intrinsically interwoven and a bi-directional association of the human-nature relationship [18, 19], and those aspects can be further described and developed in future studies with childhood cancer survivors [22]. Wilderness therapy has been developed and takes place within specific sociocultural contexts, and is influenced by those contexts and associated beliefs [18], which has an effect on participants and the stories that are shared. It could be argued that expressions of nature experience are also influenced by an overly romantic view, which informs beliefs

about how nature "ought" to be perceived. Wilderness Therapy originations in the US (and UK) may have been influenced by a 'conquering of nature' mentality, hence connecting to and taking further the supposed Frontier mentality of the American people [84].

Finally, this scoping review identified a lack of RCTs on effectiveness of wilderness programs for childhood cancer survivors. Performing RCTs is challenging. Wilderness-related programs occur in real-life contexts where randomization is often not feasible or desirable [76]. Blinding of participants to the intervention is not possible, and since participants are self-selected, they often have a strong treatment preference. Possible solutions could be to initiate pragmatic RCTs in which wilderness programs are compared to another active health promotion intervention, or an attention-control group that might be attractive for childhood cancer survivors, such as a relaxing holiday. If ethically allowed, the sharing of research questions and hypothesis with participants could be restricted; other than to tell them that two possible promising interventions are being investigated. This could potentially help to minimize expectation bias. Designs other than RCTs have also been proposed to further build the evidence base for wilderness programs, such as comparison time series, regression discontinuity, dynamic wait-listed designs, stepped wedge, and regression point displacement [23]. Quasi-experimental approaches like these may also be part of larger mixed-methods designs, including high-quality case studies and in-depth qualitative/mixed method studies which would take advantage of the inherent strengths of both quantitative and qualitative research designs.

## Conclusion

This scoping review provides valuable insights into the concept, content and outcome of wilderness programs for childhood cancer survivors that are of interest to childhood cancer survivors themselves, their families, practitioners, clinicians, and researchers. The mapping of program and participant characteristics, and gaps that were identified, provide guidance for future program development and research to further the evidentiary basis for these programs in childhood cancer survivors.

## Supporting information

**S1 File. Prisma-ScR checklist.**
(PDF)

**S2 File. Search strategy.** Results of the comprehensive search strategy in databases.
(PDF)

**S3 File. Data extraction form.**
(PDF)

**S4 File. Language excluded articles.** Possible relevant articles in other languages.
(PDF)

**S5 File. Summary article characteristics.** Summary of included article characteristics.
(PDF)

**S6 File. Health-related outcomes.** Health-related outcomes of wilderness therapy. * (n) = the number of articles in which the outcome is reported.
(PDF)

**S7 File. Methodological quality.** Assessment of the methodological quality of each included article. Assessed according to the following checklists: [1]The Joanna Briggs Institute (2019): Critical Appraisal tools for use in JBI Systematic Reviews. [2]MMAT (2019): Mixed Methods

Appraisal Tool.
(PDF)

**S1 Dataset.**
(XLSX)

## Acknowledgments

The authors are grateful to Erik Fransson and Sandra Häggander of Young Cancer Sweden (Ung Cancer: https://ungcancer.se/) for their support with data reporting, and their willingness to critically read the manuscript. The authors thank Tine Lillegård Bergli of NAFKAM (https://nafkam.no/en) for making the figures for this scoping review.

## Author Contributions

**Conceptualization:** Mats Jong, E. Anne Lown, Winnie Schats, Michelle L. Mills, Heather R. Otto, Leiv E. Gabrielsen, Miek C. Jong.

**Data curation:** Mats Jong, Winnie Schats, Michelle L. Mills, Miek C. Jong.

**Formal analysis:** Mats Jong, E. Anne Lown, Winnie Schats, Miek C. Jong.

**Investigation:** Mats Jong, E. Anne Lown, Michelle L. Mills, Heather R. Otto, Miek C. Jong.

**Methodology:** Mats Jong, E. Anne Lown, Winnie Schats, Leiv E. Gabrielsen, Miek C. Jong.

**Project administration:** Miek C. Jong.

**Validation:** Heather R. Otto, Leiv E. Gabrielsen, Miek C. Jong.

**Writing – original draft:** Miek C. Jong.

**Writing – review & editing:** Mats Jong, E. Anne Lown, Winnie Schats, Michelle L. Mills, Heather R. Otto, Leiv E. Gabrielsen, Miek C. Jong.

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
