## [Decision Letter · Decision Letter 0]

8 Sep 2020

PONE-D-20-09939

A Scoping Review to Map the Concept, Content, and Outcome of Wilderness Programs for Childhood Cancer Survivors

PLOS ONE

Dear Dr. Jong,

Thank you for submitting your manuscript to PLOS ONE. After careful consideration, we feel that it has merit but does not fully meet PLOS ONE’s publication criteria as it currently stands. Therefore, we invite you to submit a revised version of the manuscript that addresses the points raised during the review process.

We look forward to receiving your revised manuscript.

Kind regards,

Lisa Susan Wieland

Academic Editor

PLOS ONE

Journal Requirements:

Reviewers' comments:

Reviewer's Responses to Questions

**Comments to the Author**

1. Is the manuscript technically sound, and do the data support the conclusions?

Reviewer #1: Partly

Reviewer #2: Yes

Reviewer #3: Yes

2. Has the statistical analysis been performed appropriately and rigorously? 

Reviewer #1: N/A

Reviewer #2: N/A

Reviewer #3: N/A

3. Have the authors made all data underlying the findings in their manuscript fully available?

Reviewer #1: Yes

Reviewer #2: Yes

Reviewer #3: No

4. Is the manuscript presented in an intelligible fashion and written in standard English?

Reviewer #1: Yes

Reviewer #2: Yes

Reviewer #3: No

5. Review Comments to the Author

Reviewer #1: Summary

Overall, this is an interesting and important scoping review, which seeks to map and discuss the concept, content and outcome of wilderness programmes for childhood cancer survivors. I have a few suggestions for the authors to consider for further clarifying/strengthening aspects of the work but I appreciate the huge amount of work that has already gone into the review so far!

Abstract

This is well written and seems to cover the key points. One point to consider... later in the review, you articulate very clearly the substantive challenges in conducting meaningful RCTs around such programmes but this critical level of reflection is lost in how RCTs are mentioned in the abstract. I appreciate the word count is limited in an abstract, but when identifying the lack of RCTs as a gap, it would be useful to acknowledge the significant challenges in conducting meaningful RCTs in this area.

Introduction

The main point to raise about the introduction, which also has implications for later content, concerns the definition being used of wilderness programmes in the review. Lots of definitions of wilderness therapy are listed on p6. Then, p7 explains that the definition used to inform this review is: ‘programs that took place in a nature setting, and where the presence of nature had a therapeutic intention’. If this were the case, the review might also have included programmes in all sorts of nature settings including those linked to horticultural therapy programmes, green care, animal assisted therapy (e.g. on care farms) etc. It doesn’t seem from the searches that this was the case and so this definition seems a little misleading. ‘Wilderness’ is a relative concept and so by conflating wilderness with nature settings in this definition, it isn’t that clear that gardens, care farms etc were not part of the focus. Some clarification of this might be helpful.

A couple of very minor points:

P5, line 65, could probably just say ‘physical’ rather than ‘physical organ-related’.

P5, line 71, should it say ‘facilitation of supportive environments for health’ or does it refer to other forms of support?

Methods

The review methods and RQs are explained well, and the supplementary materials are helpful and appropriate. Figure 1 helps to understand more about the decisions driving the selection of studies. Again, I think clarifying the issue above about the scope of the nature settings considered as ‘wilderness’ will help to understand why so many studies (1744) were excluded at the outset through not referring to ‘wilderness’ therapy. On page 12, when discussing the quality appraisal process, it might just be worth clarifying that the quality appraisal outcomes were not used to exclude further studies, just to describe the quality of those included.

It’s great that patient representatives of Young Cancer Sweden were able to participate in the reporting of results. Were they also involved in designing the review protocol and/or in any of the analytical activities?

Results

The results are well described and explicitly linked to each of the review RQs, which works well.

A key point is made on p15 about the lack of detail given around the theoretical concepts behind the programmes, and the limited consideration of the specific roles of nature within them. Given this finding, I wonder if one of the review recommendations – rather than jumping straight to the recommendation of conducting more RCTS (geared at assessing the overall effectiveness of different interventions) – is the need for more critical, in-depth qualitative/mixed method case study work to build and deepen this conceptual understanding of how and why different wilderness encounters might be influencing different aspects of health and wellbeing for childhood cancer survivors, taking care to also situate these experiences in their varied socio-cultural life circumstances. Many of the nature experiences described on p16 – e.g. feelings of simplicity, freedom, slowing down – reflect somewhat engrained Romantic nature ideals, which suggests potentially a strong cultural influence in how these programmes are framed and experienced, which is not really acknowledged.

Linked to this, when reflecting on the disproportionate number of White participants across the programmes studied (which you raise in more detail on p22), it might be useful to read work by Sarah Jaquette Ray amongst others (e.g. a useful starting point would be: https://journals.sagepub.com/doi/10.1177/0193723509338863).

I also wonder if useful to comment on how the %female participants identified across the studies maps onto the relative prevalence of childhood cancers in male/female children?

P16, line 311, mentions the category, ‘free time/leisure activities such as using mobile devices’ – I wonder how this is really linked to wilderness in the studies reviewed. Presumably can use mobile devices any time anywhere (or perhaps not in more remote settings without signal)?

P18, lines 354-356, suggests health-related outcomes around discomfort, psychological distress and alienation decreased upon participation in the wilderness programmes. I wonder if the studies reviewed explain when the baseline measures were taken. E.g. could it be that participants were anxious, distressed etc as a result of the prospect of taking part in an unfamiliar wilderness programme, and that this decreased once the programme was underway and they started to know what to expect/acclimatise – or is it very clear from the studies that the anxiety, distress, alienation was already there e.g. as a result of their everyday experiences of life with/after cancer?

P19, lines 366-375, would be great to give an indication of the timeframes over which this transference was measured in the studies. Lines 374-375 suggest we don’t know how this transference affects health over the ‘longer term’, but what was the shorter time frame over which it was identified by the studies reviewed?

P20, Table 2, minor point about the last line, does ‘current medication/therapies’ mean %of participants still taking medication/having therapies vs those who weren’t, or something else?

P20-21, RQ 8 responses – great to reflect on how participation is funded. At some stage, either here or later in the discussion (e.g. around p25), it might be worth making the point that if schemes rely on private funding, there are potential implications for income-related health inequalities.

P21, RQ 10 responses - although apparent from S8, it might also be useful to give a brief summary of the main quality issues that were identified across the studies here (especially given the number of quality issues seemingly identified across the mixed methods studies).

P22, when flagging the lack of RCTs on effectiveness as a key gap in the literature, it might be worth signposting your later (very pertinent) critical reflections about why this might be and the feasibility of conducting meaningful/relevant RCTs in this area.

Discussion

This is well written and flags important issues around what studies are and are not yet measuring/capturing/reporting appropriately. Some of the critical reflections I’ve noted above could be interwoven in this section if more appropriate.

When discussing implications for practice on p26, I wonder if there’s also a discussion to be had about the risks of over-medicalising these experiences (particularly given, as you say, some children might feel therapy fatigue). Some of these medicalisation debates have been set out in relation to nature more generally here: https://www.sciencedirect.com/science/article/abs/pii/S0277953606006022?via%3Dihub and https://www.sciencedirect.com/science/article/pii/S0169204613000388 I wouldn’t expect a detailed discussion of this as I appreciate the word count is limited, just flagging the paper links in case useful in providing context!

Overall, this is a very interesting review and I hope the suggestions made will help to strengthen it further!

Reviewer #2: This is a through and well written review. My only comment would be the need to better address the difference between the idea of wilderness program vs. a camp. Most of the camps for children with cancer fit into the criteria you describe as a wilderness program so please address this in the introduction. Specifically, please address how this review is different than the many systematic reviews that have been completed on camp programs for children with cancer that all have a nature/adventure part of their programming.

Reviewer #3: Thanks for the opportunity reviewing the manuscript entitled “A Scoping Review to Map the Concept, Content, and Outcome of Wilderness Programs for Childhood Cancer Survivors”. I find your paper overall valuable and informative to read. I do have several comments before I would recommend for publication. Please carefully consider all comments and revise accordingly.

- Your literature review section provided a rather comprehensive overview of relevant existing systematic review and meta-analysis studies. One thing missing here is how existing literature has been discussing the different use in terminology, i.e., adventure-based therapy, nature-based therapy, wilderness program/therapy, and others. Although no one looked at this topic in childhood cancer survivors, but the discussions/debates regarding the constructs/concepts outside the field of childhood cancer intervention should also be an important guide to your current study. More importantly, to justify why you choose to focus on wilderness program and exclude certain other interventions. And how do you define wilderness programs.

- Also indicate how your operationalization is similar to or different from existing ways of operationalization and/or definition of similar constructs.

- Your underlying aim “to inform … the development, optimization, utilization and evidence-base of wilderness programs” seems difficult to achieve given it is a scoping review and you stated that the purpose is to scope a body of literature.

- Line 140, when you say, i.e., did you mean by “e.g.”?

- Define “cancer survivor”, does your operationalization of “survivor” include individuals who are receiving active cancer treatment? Why or why not.

- Your second inclusion criteria, when you say contextual, what do you mean? That it needs to occur in a nature setting?

- What are your exclusion criteria?

- Did you only use MsH term? Also for those non-medical data bases?

- Disagreements in classification of articles were solved by discussion between the two reviewers. What if there were disagreements that cannot be resolved?

- For your key search term, why not include both wilderness program and wilderness therapy? I don’t see a rationale of you have to pick one versus another.

- Your review and inclusion of theoretical framework for wilderness programs. It is important to mention possible theories that mainstream wilderness programs are relying on in your literature view section.

- Line 302-303 … reported nature/wilderness as the environment in which their program took place, most did not elaborate further on the role of nature within the program. This reads in conflict with your inclusion criteria which state it needs to have contextual and therapeutic intention. What you are saying is that most stated nature is the environment of the program but no further elaboration on the therapeutic intention? How would you then determine if a study should be included or not?

- When you say young cancer survivors, line 376. What do you mean by young? People who had childhood cancer and now are young? I was not sure if by young cancer survivor you meant by childhood cancer survivor.

- Line 386 “the articles included described at least n=1383 young cancer survivors” – this sentence has grammatic error and what do you mean by at least? You are not sure the total n based on the number of participants reported? I am confused.

- I disagree your use of term “disability” only in your review question 9. I think you were referring to a set of side effects, co-morbidities, and other symptoms related to cancer and cancer treatment that may or may not lead to disability. I don’t think disability is the right term to cover everything.

- Key information missing which has to do with reporting the design of included studies. Please include this in the result section.

- While your review question 11 is relevant, many gaps you identified in the literature gap are actually the very reason that you would do a scoping review but rather than a systematic review. In other words, if there are RCTs, studies including long term effects, then a scoping review would not be appropriate here. So, I don’t think the first two gaps you mention should be in your result section, but rather, in your literature review section. You can word it differently in your review question, looking at published studies and anticipating limited RCTs and long term follow up, then justifying you are doing a scoping review. I think the first two bullet points for review question 11 was underwhelming given you are doing a scoping review.

- You included different languages, and I assume studies from different countries. Did you look at program differences that are published in different studies?

6. PLOS authors have the option to publish the peer review history of their article (what does this mean?). If published, this will include your full peer review and any attached files.

Reviewer #1: No

Reviewer #2: No

Reviewer #3: **Yes: **Anao Zhang

---

## [Author Response · Author response to Decision Letter 0]

13 Oct 2020

We have now made major revisions to the manuscript according to the comments of the reviewers and are of the opinion that our manuscript has greatly improved. 

First, a general point that we would like to address is that several questions of the reviewers below concern the request for information that has already been published in the protocol of this scoping review (Jong et al. BMJ Open, 2019). Since the detailed methodology of this scoping review has been published, and our published protocol is clearly referenced in the current manuscript, we tried to find a better balance repeating only some of the most important methodologic details in the present manuscript.

Hereby we give a point-by-point response to the comments:

Reviewer 1: Overall, this is an interesting and important scoping review, which seeks to map and discuss the concept, content and outcome of wilderness programmes for childhood cancer survivors. I have a few suggestions for the authors to consider for further clarifying/strengthening aspects of the work but I appreciate the huge amount of work that has already gone into the review so far!

Answer: We thank the reviewer for the overall evaluation that our scoping review is of interest and importance, and for the valuable suggestions in order to further clarify and strengthen our manuscript.

1.1 Abstract. This is well written and seems to cover the key points. One point to consider... later in the review, you articulate very clearly the substantive challenges in conducting meaningful RCTs around such programmes but this critical level of reflection is lost in how RCTs are mentioned in the abstract. I appreciate the word count is limited in an abstract, but when identifying the lack of RCTs as a gap, it would be useful to acknowledge the significant challenges in conducting meaningful RCTs in this area.

Answer: We now acknowledge the challenges in conducting RCTs in the conclusion section of the abstract. However, in order to remain within the 300 word-limitation, we have made the decision to delete detailed information on databases that were searched.

1.2 Introduction. The main point to raise about the introduction, which also has implications for later content, concerns the definition being used of wilderness programmes in the review. Lots of definitions of wilderness therapy are listed on p6. Then, p7 explains that the definition used to inform this review is: ‘programs that took place in a nature setting, and where the presence of nature had a therapeutic intention’. If this were the case, the review might also have included programmes in all sorts of nature settings including those linked to horticultural therapy programmes, green care, animal assisted therapy (e.g. on care farms) etc. It doesn’t seem from the searches that this was the case and so this definition seems a little misleading. ‘Wilderness’ is a relative concept and so by conflating wilderness with nature settings in this definition, it isn’t that clear that gardens, care farms etc were not part of the focus. Some clarification of this might be helpful.

Answer: We thank the reviewer for this critical remark and the possibility to further clarify how we operationalized wilderness programs in our scoping review. Upon our intention to explore the concept, content, and outcome of wilderness programs for childhood cancer survivors in a scoping review, we were directly confronted with the dilemma how to define wilderness programs. In current literature, wilderness programs are identified interchangeably with numerous other terms such as wilderness therapy, adventure-based therapy, nature therapy, therapeutic camping, adventure camp, and challenge courses. Along this large variance in use of terms, there are multiple definitions of wilderness therapy. We now more clearly describe this dilemma in the Introduction of our revised manuscript, and that the different use in terminology and definitions reflect the historical and socio-cultural contexts and traditions in which wilderness programs have developed. 

Because it was not possible to find a consistent and well-defined definition of wilderness programs/therapy, we predefined a pragmatic operational concept of wilderness programs for the purpose of our scoping review. The predefined operational concept is described in the protocol of our scoping review that was published last year in BMJ Open (Jong et al, 2019): “Wilderness-related therapies such as adventure therapy, recreation programmes, nature-based programmes, outdoor programmes, open-air programmes, forest bathing and bush-craft, in which the role of nature has both a contextual and therapeutic premise”. We further described in the published protocol “studies focusing on related topics that do not primarily evaluate wilderness and/or nature experiences, such as evaluation of hospital gardens, physical exercise programmes, and animal-assisted therapy, or which do not explicitly offer a program (individuals spending time hiking or star gazing on their own) will be excluded in the scoping review”. In the Methodology section of the revised manuscript we have now added this exclusion criterion, in line with the previously published protocol. Additionally, in the Introduction section of our revised manuscript we now further explain what is meant by inclusion of the phrase “the role of nature should have both a contextual and therapeutic premise”. 

1.3 P5, line 65, could probably just say ‘physical’ rather than ‘physical organ-related’. 

Answer: We agree with the reviewer and have revised the text accordingly.

1.4 P5, line 71, should it say ‘facilitation of supportive environments for health’ or does it refer to other forms of support?

Answer: We agree with the reviewer and have revised the text accordingly to facilitation of supportive environments for health.

1.5 Methods

The review methods and RQs are explained well, and the supplementary materials are helpful and appropriate. Figure 1 helps to understand more about the decisions driving the selection of studies. Again, I think clarifying the issue above about the scope of the nature settings considered as ‘wilderness’ will help to understand why so many studies (1744) were excluded at the outset through not referring to ‘wilderness’ therapy. 

Answer: See also our answer to point 1.2 above. We have now further clarified our predefined operationalization of the concept of wilderness programs and the role of nature, as well as included the exclusion criteria for other type of programmes in the Methodology section thereby referring to the previously published protocol of the scoping review.

1.6 On page 12, when discussing the quality appraisal process, it might just be worth clarifying that the quality appraisal outcomes were not used to exclude further studies, just to describe the quality of those included.

Answer: We have now revised the text in the Methodology section as suggested in order to clarify the fact that quality appraisal was not performed to exclude studies for further analysis in the systematic review, but rather to describe the quality of those studies included so as to potentially identify gaps in literature.

1.7 It’s great that patient representatives of Young Cancer Sweden were able to participate in the reporting of results. Were they also involved in designing the review protocol and/or in any of the analytical activities?

Answer: As previously published (Jong et al, BMJ Open 2019), patient representatives were not involved in designing of the scoping review protocol and/or in any of the analytical activities. We are grateful to Young Cancer Sweden for their support with data reporting, and their willingness to critically read the manuscript of this scoping review. Furthermore, we are happy to inform the reviewer that based on the outcome of this scoping review, patient representatives of Young Cancer Sweden are involved in the design and process of performing a randomized controlled study on the effects of a wilderness program for adolescent and young adult cancers survivors in Sweden (The WAYA study: https://www.miun.se/en/Research/research-projects/ongoing-research-projects/waya/)

1.8 Results. The results are well described and explicitly linked to each of the review RQs, which works well. A key point is made on p15 about the lack of detail given around the theoretical concepts behind the programmes, and the limited consideration of the specific roles of nature within them. Given this finding, I wonder if one of the review recommendations – rather than jumping straight to the recommendation of conducting more RCTS (geared at assessing the overall effectiveness of different interventions) – is the need for more critical, in-depth qualitative/mixed method case study work to build and deepen this conceptual understanding of how and why different wilderness encounters might be influencing different aspects of health and wellbeing for childhood cancer survivors, taking care to also situate these experiences in their varied socio-cultural life circumstances. Many of the nature experiences described on p16 – e.g. feelings of simplicity, freedom, slowing down – reflect somewhat engrained Romantic nature ideals, which suggests potentially a strong cultural influence in how these programmes are framed and experienced, which is not really acknowledged. Linked to this, when reflecting on the disproportionate number of White participants across the programmes studied (which you raise in more detail on p22), it might be useful to read work by Sarah Jaquette Ray amongst others (e.g. a useful starting point would be: https://journals.sagepub.com/doi/10.1177/0193723509338863).

Answer: We thank the reviewer for referring to the work by Sarah Jaquette Ray, which we read with great interest. We are in complete agreement with the reviewer that more in-depth qualitative/mixed methods case studies are necessary to gain better insight and conceptual understanding of how and why wilderness programmes may promote health and well-being of childhood cancer survivors. This is also what we more or less meant with our statement at the end of the Discussion section (of our previously submitted manuscript), that high-quality case studies can aid our efforts to extract valid and useful inferences from our quantitative datasets. In the revised manuscript, we now more explicitly describe that case studies, and mixed methods research studies are recommended (Page 30-31, “Manuscript with changes highlighted”). Connected to this, we have also included text to lift the sociocultural aspects further, with reference to the article of Ray (page 31, “Manuscript with changes highlighted”).).

1.9 I also wonder if useful to comment on how the % female participants identified across the studies maps onto the relative prevalence of childhood cancers in male/female children?

Answer: There are some differences, with boys generally more susceptible to cancer than girls at a young age, and then the prevalence in girls overtaking that of boys later in childhood. We have now commented on the overrepresentation of females in the wilderness programs, and made reference to this in the Discussion section of the revised manuscript (Page 26, “Manuscript with changes highlighted”). 

1.10 P16, line 311, mentions the category, ‘free time/leisure activities such as using mobile devices’ – I wonder how this is really linked to wilderness in the studies reviewed. Presumably can use mobile devices any time anywhere (or perhaps not in more remote settings without signal)?

Answer: Figure 2 gives an overview of all categories and underlying activities in the included wilderness programs that we mapped. Gill et al. (2016) described the use of mobile devices as free time activities in an outdoor adventure therapy program. This program was camp-based, and participants were out in nature/wilderness for kayaking, rock climbing and surfing. The study unfortunately lacks further detailed information on how or for what these mobile devices were used. 

1.11 P18, lines 354-356, suggests health-related outcomes around discomfort, psychological distress and alienation decreased upon participation in the wilderness programmes. I wonder if the studies reviewed explain when the baseline measures were taken. E.g. could it be that participants were anxious, distressed etc as a result of the prospect of taking part in an unfamiliar wilderness programme, and that this decreased once the programme was underway and they started to know what to expect/acclimatise – or is it very clear from the studies that the anxiety, distress, alienation was already there e.g. as a result of their everyday experiences of life with/after cancer?

Answer: We now describe in the Discussion section of the revised manuscript that it seems unlikely that the observed decrease in psychological distress and discomfort as observed in the studies are the result of higher anxiety levels, discomfort and distress at baseline caused by the prospect of taking part in an unfamiliar wilderness program. From a previous systematic review of one of the co-authors of this scoping review, it is known that childhood cancer survivors have greater psychological distress as a result of their everyday experiences of life with/after cancer (Lown et al. 2015). Furthermore, in the study by Rosenberg et al. (2014), baseline measurements were performed two weeks prior to the start of the program, and outcomes were significantly decreased in the wilderness program group compared to a wait-list control group. In the study by Zebrack et al (2017), baseline measurements were also performed two weeks prior to the start of the program. Directly after participation in the program it was observed that psychological distress decreased more significantly in those with moderate to severe-distress at baseline compared to those with mild to no-distress at baseline. However, psychological distress increased again in the one-month follow-up interview after participation in the program.

In the study by Paquette et al (2017), a decreased feeling of isolation was qualitatively described, associated with a better capacity to express emotions and connect with others after participation in the program. Less discomfort was qualitatively described and linked to the ability to better deal with weather conditions after participation in the program. 

1.12 P19, lines 366-375, would be great to give an indication of the timeframes over which this transference was measured in the studies. Lines 374-375 suggest we don’t know how this transference affects health over the ‘longer term’, but what was the shorter time frame over which it was identified by the studies reviewed?

Answer: In the Results section of the revised manuscript, we have now described the timeframes over which the transference was measured in the studies. The timeframe in the study by Kessel et al. (1985) was 6 months after participation in the program, and the timeframe in the study by Wagner et al. (2014) was directly after participation in the program.

1.13 P20, Table 2, minor point about the last line, does ‘current medication/therapies’ mean % of participants still taking medication/having therapies vs those who weren’t, or something else?

Answer: Table 2 shows that 3 out of 15 studies reported whether participants were on active medication/therapies at the time of participation in the program. Out of the 345 participants that were included in these three studies, about half were on active treatment (medication/therapies). We have now clarified this in Table 2 and in the Results section of the manuscript.

1.14 P20-21, RQ 8 responses – great to reflect on how participation is funded. At some stage, either here or later in the discussion (e.g. around p25), it might be worth making the point that if schemes rely on private funding, there are potential implications for income-related health inequalities.

Answer: We thank the reviewer for being sensitive to this possible influence creating disparities. We have responded to this issue in the section in the Discussion section under the heading implication for practice, where the funding of wilderness programs is discussed saying that, when participation in wilderness programs solely relies on private funding, there is a greater likelihood that these type of programs will enrol participants with greater economic privilege and exclude those with a lower socioeconomic status from participation.

1.15 P21, RQ 10 responses - although apparent from S8, it might also be useful to give a brief summary of the main quality issues that were identified across the studies here (especially given the number of quality issues seemingly identified across the mixed methods studies).

Answer: We thank the reviewer for this good suggestion. We have now included in the Results section a brief summary description of the main quality issues identified across the 15 studies included.

1.16 P22, when flagging the lack of RCTs on effectiveness as a key gap in the literature, it might be worth signposting your later (very pertinent) critical reflections about why this might be and the feasibility of conducting meaningful/relevant RCTs in this area.

Answer: Based on this comment, we found it useful to briefly summarize the findings related to the observed gaps in literature [discussed under review question 11 in the paper] in the Results section. We have therefore added more descriptive text relating specifically to the facts as described in the included studies so as to maintain a clear distinction between reporting the findings/observations in the Results section, and discussing the possible underlying causes and implications in the Discussion section. In addition, we have now also added commentary on the lack of information on safety of wilderness programs in our discussion of the identified gaps in literature. This is in line with our recommendations under the heading Implications for Practice that more safety studies on wilderness programs are necessary.

1.17 Discussion. This is well written and flags important issues around what studies are and are not yet measuring/capturing/reporting appropriately. Some of the critical reflections I’ve noted above could be interwoven in this section if more appropriate.

Answer: We have now revised the Discussion section of the manuscript, in line with the critical reflections as suggested by the reviewer (e.g. possible high psychological distress before participation in the program, prevalence of cancer in boys versus girls, private funding of programs, high quality case studies). 

1.18 When discussing implications for practice on p26, I wonder if there’s also a discussion to be had about the risks of over-medicalising these experiences (particularly given, as you say, some children might feel therapy fatigue). Some of these medicalisation debates have been set out in relation to nature more generally here: https://www.sciencedirect.com/science/article/abs/pii/S0277953606006022?via%3Dihub and https://www.sciencedirect.com/science/article/pii/S0169204613000388 I wouldn’t expect a detailed discussion of this as I appreciate the word count is limited, just flagging the paper links in case useful in providing context!

Answer: In the Discussion section of the manuscript, we have now included these two references as suggested by the reviewer, within the context of the current trend in healthcare that nature and exercise are “medicalized” to nature on prescription and physical activity on prescription.

1.19 Overall, this is a very interesting review and I hope the suggestions made will help to strengthen it further!

Answer: We thank the reviewer for these kind words, and are of the opinion that the revisions we have made at the suggestion of the reviewer have strengthened the quality and interest of our scoping review.

Reviewer 2: This is a through and well-written review. 

2.1 My only comment would be the need to better address the difference between the idea of wilderness program vs. a camp. Most of the camps for children with cancer fit into the criteria you describe as a wilderness program so please address this in the introduction. 

Answer: With respect to this point, we would also like to refer to our answer to Reviewer 1 above (point 1.2). In the Introduction of our revised manuscript, we have now clarified our predefined operationalization of the concept of wilderness programs and the role of nature in the program, as well as included the exclusion criteria for other types of programs in the Methodology section, thereby referring to the previously published protocol of the scoping review. Our predefined concept of wilderness programs did not specifically set out to exclude cancer camps, and indeed one study on a cancer camp met the inclusion criteria of our scoping review. We have described in the Discussion section under Strengths and Limitations that all other articles on cancer camps that we found were not included in this scoping review due to the fact that they did not meet our inclusion criteria. The specific reason for study exclusion in these cases was the fact that the study did not specify the setting of the cancer camp in nature, and/or because the study did not include any information that made clear a therapeutic intent behind being in nature or nature activities. While nature is indeed often integrated into camp life, we are aware that it is a limitation of the current scoping review that lack of explicit mention of nature in articles on cancer camps may have excluded articles that could be relevant for childhood cancer survivors (see Discussion section, Strengths and Limitations heading).

2.2 Specifically, please address how this review is different than the many systematic reviews that have been completed on camp programs for children with cancer that all have a nature/adventure part of their programming.

Answer: We thank the reviewer for this critical, but constructive comment. In the Introduction section, we now refer to and describe two previous systematic reviews on cancer camps for children that have been published (Martiniuk et al. 2014, Neville et al. 2019). We specifically address in the Introduction how our scoping review is different from these, and the other systematic reviews previously published: “Although cancer camps often take place in an outdoor or wilderness setting, these and other previously published systematic reviews have not specifically addressed or investigated the role of wilderness and nature in programs for childhood cancer survivors”. 

A third systematic review on cancer camps (Kelada et al. 2020) has been published since we finished our scoping review. We have now included a reference to the findings of this third review in our Discussion section. We would also like to mention that when comparing the 15 studies that were included in our scoping review with the studies that were included in the systematic reviews on camp programmes, the difference between our review and the others is clear. Out of the 20 studies that were included in the systematic review on cancer camps by Martiniuk et al (2014), only 1 study was included in our scoping review analysis. Out of the 18 studies that were included in the systematic review on cancer camps by Neville et al (2019), just 1 overlapped with our scoping review. In the recently published systematic review on cancer camps by Kelada et al. (2020), none of the 19 articles they included were included in our scoping review. In addition, we have now also revised the Discussion section of the manuscript to increase clarity with regard to how the findings of our scoping review align with the findings of the systematic reviews on cancer camps, and also to highlight the ways in which our scoping review adds to them. 

Reviewer 3: Thanks for the opportunity reviewing the manuscript entitled “A Scoping Review to Map the Concept, Content, and Outcome of Wilderness Programs for Childhood Cancer Survivors”. I find your paper overall valuable and informative to read. I do have several comments before I would recommend for publication. Please carefully consider all comments and revise accordingly.

Answer: We thank the Reviewer for his overall evaluation that our paper is valuable and informative, and for the constructive comments that allowed us the opportunity to improve and strengthen our manuscript. 

3.1 Your literature review section provided a rather comprehensive overview of relevant existing systematic review and meta-analysis studies. One thing missing here is how existing literature has been discussing the different use in terminology, i.e., adventure-based therapy, nature-based therapy, wilderness program/therapy, and others. Although no one looked at this topic in childhood cancer survivors, but the discussions/debates regarding the constructs/concepts outside the field of childhood cancer intervention should also be an important guide to your current study. More importantly, to justify why you choose to focus on wilderness program and exclude certain other interventions. And how do you define wilderness programs.

Answer: With respect to this point, we would also like to refer to our answer to Reviewer 1 above (point 1.2). In the Introduction of our revised manuscript, we have now clarified our predefined operationalization of the concept of wilderness programs and the role of nature in the program, as well as included the exclusion criteria for other types of programs in the Methodology section. These changes now reference the description in our previously published protocol for the scoping review. In addition, we now describe more clearly the way the different uses of terminology and variable definitions of wilderness therapy reflect the historical and socio-cultural contexts and traditions from which wilderness programmes have developed.

3.2 Also indicate how your operationalization is similar to or different from existing ways of operationalization and/or definition of similar constructs.

Answer: We now describe in the Introduction section of the revised manuscript that in the absence of a consistent and universally accepted definition, a pragmatic operational concept of wilderness programmes was predefined for the purpose of this scoping review (with reference to the previously published protocol). Furthermore, we now describe what is meant by our requirement that the role of nature should have both a contextual and therapeutic premise, by which we are referring to the role of nature in the wilderness therapy construct as previously published by one of the co-authors (Harper, Gabrielsen et al. 2018).

3.3 Your underlying aim “to inform … the development, optimization, utilization and evidence-base of wilderness programs” seems difficult to achieve given it is a scoping review and you stated that the purpose is to scope a body of literature.

Answer: We have now revised this in the Introduction section of the revised manuscript to clarify the aim of this scoping review. The aim is to systematically map the concept, content and outcome of wilderness programs for childhood cancer survivors, with the purpose of informing childhood cancer survivors, their families, practitioners, clinicians, and researchers.

The section Implication for Practice in the Discussion of this manuscript describes how the outcome of this scoping review informs childhood cancer survivors, their families, practitioners, clinicians and researchers. In this section, recommendations are made for further development, optimization, utilization and evaluation of wilderness programs.

3.4 Line 140, when you say, i.e., did you mean by “e.g.”?

Answer: We thank the reviewer for this sharp observation, indeed it should be e.g. instead of i.e. 

We have revised the text accordingly.

3.5 Define “cancer survivor”, does your operationalization of “survivor” include individuals who are receiving active cancer treatment? Why or why not.

Answer: The term “childhood cancer survivors” is defined in the protocol of this scoping review that was published in BMJ Open (2019): Describing childhood cancer survivors, meaning participants

of any sex diagnosed with cancer before the age of 21. A person is defined as a cancer survivor from

the moment of cancer diagnosis throughout life. Thus, this also includes individuals who are receiving active cancer treatment. We have now included the definition of childhood cancer survivors in the Methods section of the revised manuscript.

3.6 Your second inclusion criteria, when you say contextual, what do you mean? That it needs to occur in a nature setting?

Answer: With respect to the role of nature, we now explain in the Introduction of the revised manuscript that a wilderness program should take place in nature, and that nature-related activities within the program are intended to have a therapeutic effect.

3.7 What are your exclusion criteria?

Answer: Exclusion criteria are described in the protocol of this scoping review, previously published in BMJ Open (Jong et al, 2019). We have now included the exclusion criteria in the Methods section of the revised manuscript: “Studies that lacked one or two of these items were excluded. Studies focusing on related topics that did not primarily evaluate wilderness and/or nature experiences, such as evaluation of hospitals gardens, physical exercise programs and animal-assisted therapy, or which did not explicitly offer a program (individuals spending time hiking or star gazing on their own) were also excluded from this scoping review”.

3.8 Did you only use MeSH term? Also for those non-medical data bases?

Answer: For all databases we used both MeSH terms (or comparable controlled vocabularies) and free text terms, usually terms in title and/or abstract of the publications. In databases without controlled vocabularies where only a simplified search strategy could be used we translated the search strategy as broadly as possible to have maximum search yield. We have now explained this more clearly in the Methods section of the revised manuscript.

3.9 Disagreements in classification of articles were solved by discussion between the two reviewers. What if there were disagreements that cannot be resolved?

Answer: In the published protocol of this scoping review (Jong et al. BMJ Open 2019), we have written in the Methods section under the heading Study Selection: “Disagreement between the two reviewers will be discussed with a third author (EAL), and final decisions will be made”. However, in the process of classification all initial disagreements were solved in further discussion between the two reviewers and consulting a third author was not necessary. We have now revised the text into: “Initial disagreements in classification of articles were resolved by further discussion between the two reviewers, obviating the need for consultation with a third author”.

3.10 For your key search term, why not include both wilderness program and wilderness therapy?

I don’t see a rationale of you have to pick one versus another.

Answer: We hereby confirm that we have included both wilderness program and wilderness therapy in our search strategies, and also other terms like wilderness treatment and wilderness training. In the Methods section of our manuscript, we make reference to the published protocol (BMJ Open, 2019) in which the key search terms for this scoping review are listed in Box 2. 

By using a proximity/adjacency operator we combined 'wilderness', 'adventure', 'nature' and similar terms with 'program', 'therapy', 'treatment' and so on. That way we could include not only 'wilderness therapy' and 'wilderness program' but also terms where 'wilderness' and 'therapy' were not directly next to each other, like in 'wilderness adventure therapy'.

3.11 Your review and inclusion of theoretical framework for wilderness programs. It is important to mention possible theories that mainstream wilderness programs are relying on in your literature view section.

Answer: We thank the reviewer for this good suggestion. We have now included a summary of the most important theories underlying wilderness programs in the Introduction section of the revised manuscript.

3.12 Line 302-303 … reported nature/wilderness as the environment in which their program took place, most did not elaborate further on the role of nature within the program. This reads in conflict with your inclusion criteria which state it needs to have contextual and therapeutic intention. What you are saying is that most stated nature is the environment of the program but no further elaboration on the therapeutic intention? How would you then determine if a study should be included or not?

Answer: We thank the reviewer for this critical comment and understand the confusion. We have now revised the text accordingly in order to clarify the fact that all articles included met the inclusion criteria: “While all articles reported nature/wilderness as the environment in which their program took place, and had included some kind of description from which it could be concluded that nature had a therapeutic intention in the program, most did not describe in much detail on the role of nature within their program”.

3.13 When you say young cancer survivors, line 376. What do you mean by young? People who had childhood cancer and now are young? I was not sure if by young cancer survivor you meant by childhood cancer survivor.

Answer: We agree that the term “young cancer survivors” is not sufficiently clarified in the manuscript. The reference to “young cancer survivors” in the Results section, rather than childhood cancer survivors, was because we included four articles in the scoping review on young adult cancer survivors [27, 38, 41, 44], where the age at cancer diagnosis was not specified in the studies. It is thus not certain that the diagnosis of participants in these studies occurred before the age of 21, and thus it is not certain whether the participants in these studies comply with the definition of childhood cancer survivors. In order to avoid possible confusion, we have now consistently revised “young cancer survivors” to read “childhood cancer survivors”, in line with the aim of this scoping review. Furthermore, it is clearly described in the Discussion section under the heading Strengths and Limitations section that the results reported in these four articles may largely or partly concern young adult survivors, and not solely childhood cancer survivors. 

3.14 Line 386 “the articles included described at least n=1383 young cancer survivors” – this sentence has grammatic error and what do you mean by at least? You are not sure the total n based on the number of participants reported? I am confused.

Answer: In three studies (Carlsson 2007, Epstein 2004 and Pearson 1989), the number of participants was not described. That is the reason why we wrote that at least n=1383 young cancer survivors were studied in the 15 included studies. However, in order to avoid any confusion, we have now revised this sentence, since we clearly make reference to the 12 studies that have reported results on a total of 1383 cancer survivors: “ The articles described young cancer survivors (n=1383) with a broad age range from age 8–40 years (n=1350) [27, 33, 35, 36, 38-41, 43, 44]”.

3.15 I disagree your use of term “disability” only in your review question 9. I think you were referring to a set of side effects, co-morbidities, and other symptoms related to cancer and cancer treatment that may or may not lead to disability. I don’t think disability is the right term to cover everything.

Answer: We agree with the reviewer that the term “disability” does not sufficiently capture possible impairments, disabilities, and late effects either of the cancer itself, or of cancer-related treatment. We have now revised the text in the Results section under review question 9 accordingly: “Participation in wilderness programs was possible for childhood cancer survivors with various comorbidities and/or disabilities, impairments or late health effects as caused either by the cancer itself or by cancer-related treatment”.

3.16 Key information missing which has to do with reporting the design of included studies. Please include this in the result section.

Answer: The design of the included studies is described in the Results section of the manuscript under the heading Characteristics of Studies: “Included articles were methodologically diverse, consisting of text & opinion papers (n=5) [31-34, 37], qualitative studies (n=4) [39-42], studies using a mixed-methods design (n=3) [35, 36, 43], or quasi-experimental studies (n=3). Further detailed information on the study design of each included study is listed in the Table that is included as supporting information S6”.

3.17 While your review question 11 is relevant, many gaps you identified in the literature gap are actually the very reason that you would do a scoping review but rather than a systematic review. In other words, if there are RCTs, studies including long term effects, then a scoping review would not be appropriate here. So, I don’t think the first two gaps you mention should be in your result section, but rather, in your literature review section. You can word it differently in your review question, looking at published studies and anticipating limited RCTs and long term follow up, then justifying you are doing a scoping review. I think the first two bullet points for review question 11 was underwhelming given you are doing a scoping review.

Answer: Our review question 11, and subsequent findings that there is a lack of RCTs and long-term effect studies on this topic are very well in line with the aims and expected outcomes of a scoping review. We hereby refer to four articles that describe the definition and aims of systematically performed scoping reviews: 

“A scoping review or scoping study is a form of knowledge synthesis that addresses an exploratory research question aimed at mapping key concepts, types of evidence, and gaps in research related to a defined area or field by systematically searching, selecting, and synthesizing existing knowledge”

(Colquhoun et al. J Clin Epidem, 2014).

In summary, the following aims of a scoping review have been described:

1. To examine the extent, range and nature or research activities

2. To determine the value of undertaking a systematic review

3. To summarize and disseminate research findings

4. To identify gaps in the existing literature

(Arksey & O'Malley, Intern J Soc Res Meth, 2005; Peters et al., Int J Evid Based Healthc 2015; Munn et al, BMC Med Res Meth, 2018)

3.18 You included different languages, and I assume studies from different countries. Did you look at program differences that are published in different studies?

Answer: In the Results, under the heading Study Characteristics we have described the number of different countries from which the studies originated: “Most articles originated in the USA (n=10) [27, 31-33, 35, 37, 38, 41, 42, 44], others were from Canada (n=3)[34, 36, 40], the UK (n=1) [39], and New Zealand (n=1) [43]. Program differences between the different studies are listed in Figure 2, and in more detail in the Table that is included as supporting information S6.

Additional requirements of the journal:

Answer: We have revised the manuscript so that it meets the style requirements of PLOS ONE.

2. Please include captions for your Supporting Information files at the end of your manuscript, and update any in-text citations to match accordingly. 

Answer: We have now included captions for our supporting information files at the end of the manuscript and updated in-text citations accordingly.

---

## [Decision Letter · Decision Letter 1]

1 Dec 2020

A Scoping Review to Map the Concept, Content, and Outcome of Wilderness Programs for Childhood Cancer Survivors

PONE-D-20-09939R1

Dear Dr. Jong,

We’re pleased to inform you that your manuscript has been judged scientifically suitable for publication and will be formally accepted for publication once it meets all outstanding technical requirements.

Kind regards,

Lisa Susan Wieland

Academic Editor

PLOS ONE

Additional Editor Comments (optional):

Reviewers' comments:

Reviewer's Responses to Questions

**Comments to the Author**

1. If the authors have adequately addressed your comments raised in a previous round of review and you feel that this manuscript is now acceptable for publication, you may indicate that here to bypass the “Comments to the Author” section, enter your conflict of interest statement in the “Confidential to Editor” section, and submit your "Accept" recommendation.

Reviewer #1: All comments have been addressed

Reviewer #2: All comments have been addressed

Reviewer #3: All comments have been addressed

2. Is the manuscript technically sound, and do the data support the conclusions?

Reviewer #1: Yes

Reviewer #2: Yes

Reviewer #3: (No Response)

3. Has the statistical analysis been performed appropriately and rigorously? 

Reviewer #1: N/A

Reviewer #2: N/A

Reviewer #3: (No Response)

4. Have the authors made all data underlying the findings in their manuscript fully available?

Reviewer #1: Yes

Reviewer #2: Yes

Reviewer #3: (No Response)

5. Is the manuscript presented in an intelligible fashion and written in standard English?

Reviewer #1: Yes

Reviewer #2: Yes

Reviewer #3: (No Response)

6. Review Comments to the Author

Reviewer #1: The authors have provided a thoughtful and thorough response to my previous review comments, and I look forward to (hopefully!) seeing this important paper published. Well done!

Reviewer #2: (No Response)

Reviewer #3: (No Response)

7. PLOS authors have the option to publish the peer review history of their article (what does this mean?). If published, this will include your full peer review and any attached files.

Reviewer #1: No

Reviewer #2: No

Reviewer #3: No

---

## [Editor Report · Acceptance letter]

4 Dec 2020

PONE-D-20-09939R1 

A Scoping Review to Map the Concept, Content, and Outcome of Wilderness Programs for Childhood Cancer Survivors. 

Dear Dr. Jong:

I'm pleased to inform you that your manuscript has been deemed suitable for publication in PLOS ONE. Congratulations! Your manuscript is now with our production department. 

Kind regards, 

on behalf of

Dr. Lisa Susan Wieland 

Academic Editor

PLOS ONE